# Relationship between Health Status and Daily Activities Based on Housing Type among Suburban Residents during COVID-19 Self-Isolation

**DOI:** 10.3390/ijerph20032639

**Published:** 2023-02-01

**Authors:** Yangcheng Gu, Haruka Kato, Daisuke Matsushita

**Affiliations:** Department of Living Environment Design, Graduate School of Human Life and Ecology, Osaka Metropolitan University, Osaka 5588585, Japan

**Keywords:** COVID-19, SF-12v2, daily activities, suburban, detached house, apartment

## Abstract

COVID-19 significantly impacted residents’ health status and daily activities in suburban residential areas. This study elucidated the relationship between health scores, daily activities, and housing types. The method was a questionnaire survey of 378 residents of suburban residential estates in Teraikedai, Kongo District, Japan, during the COVID-19 self-isolation period. Since the survey cohort was New Town, the suburban residential area identified by the Ministry of Land, Infrastructure, Transport and Tourism was targeted. The questions included participant demographics, the Basic Survey on Japanese Social Life, and the SF-12v2. The Tukey–Kramer HSD test and stepwise decreasing logistic regression were used for the statistical analysis of the responses. The COVID-19 self-isolation led to lower physical and mental health scores than usual, and the health scores of residents living in detached houses were better than those of residents in apartments, both those over the age of 65 and those under the age of 65. There was also a correlation between residents’ daily activities and their health scores. For those aged under 65 years, the health scores of residents living in detached houses were significantly better than those living in apartments, indicating that daily activities such as sports and recreational hobbies may contribute to health scores.

## 1. Introduction

The background of this study is the deterioration in health status caused by the COVID-19 pandemic. On 11 March 2020, the World Health Organization (WHO) declared an outbreak of COVID-19. As of 7 March 2022, there were 5,720,394 confirmed cases of COVID-19 in Japan as a whole, of which 26,029 were associated deaths [1], and this number may increase further. Studies on coronaviruses show that self-isolation can reduce transmission [2]. On 7 April 2020, the Japanese government issued the first emergency declaration and set a target of reducing commuting by 40% and human contact by 30%, while encouraging self-regulation by residents [3]. In preventing COVID-19 infections, self-isolation meant that residents could not leave their homes unless necessary, so as to maintain their livelihoods. In 2022, the Japanese Cabinet recommended that Japanese residents should wear masks outside. Japanese residents were more likely to avoid infection through self-restraint. Thus, government restrictions gradually weakened. However, residents still restricted their daily activities so as to prevent infection. Compared to other countries, Japan was characterized by the residents’ self-isolation. Affected by the self-isolation, people’s range of activities was reduced by about half [4].

The study’s main research question was to determine whether the relationship between changes in daily activities and health status during COVID-19 self-isolation depended not only on age but also on housing type. The housing types in which individuals lived during the COVID-19 pandemic were more likely to cause severe effects in the elderly and cause more significant changes in daily activities. Therefore, daily activities and health status changes have been studied on an age-specific basis [5,6,7,8,9,10,11,12,13,14,15,16]. On the other hand, in studies in the field of public health before the coronavirus outbreak, the relationship between daily activities and health status was studied mainly on the basis of the residential environment [17,18]. This is because the living environment has a significant impact on health. The authors believed that the relationship between changes in daily activities and health status during the COVID-19 self-isolation period should be elucidated based on housing type as well as age.

This study aimed to elucidate the relationship between residents’ daily activities and their health status during the COVID-19 self-isolation period based on age and housing type. This study was an exhaustive questionnaire survey focusing on residents who are associated with neighborhood associations in Teraikedai 1-chome and 5-chome. The survey area was selected as Teraikedai, Tondabayashi, Osaka, as a suburban residential area with an elderly population in Japan. An exhaustive questionnaire survey of all household residents who were associated with neighborhood associations was used to investigate the relationship between nineteen daily activities and health status. That is because Japan’s most miniature community scale is the neighborhood associations scale. In the Kongo District, neighborhood association is the basis for local communities, welfare, childcare, and disaster prevision. Therefore, in Japan, neighborhood associations play an essential role in institutional decision-making regarding urban planning [19]. In addition, especially in the 1-chome and 5-chome, the neighborhood associations cover most of the detached houses and apartment residents. The above reasons indicate the validity of the analysis based on neighborhood associations. The result was expected to reveal insights into the daily activities of residents of each housing type for maintaining health.

Several studies abroad have shown that a voluntary curfew following COVID-19 self-isolation led to a significant reduction in physical activity, negatively affecting the health status of individuals. Health problems have been reported due to reduced physical activity (walking; light, moderate, and strenuous exercise) and increased sedentary time from 5 to 8 h [5]. According to the 2020 WHO physical activity advice, to maintain good health, adults should undertake 150–300 min of moderate-intensity exercise per week or 75–150 min of high-intensity exercise per week [6]. In Hagino in Osaka, it was found that 30% of the elderly had reduced physical activity, and their health had deteriorated [7]. As well as physical activities, other daily activities have also been reported to be significantly affected by this. For example, in Chinese studies, residents’ home quarantine due to the novel coronavirus increased their sleeping hours [8]. In addition, it was found that many people developed sleep disorders [9,10]. On the other hand, it was found that while diet quality and eating habits improved during the COVID-19 pandemic, increased snacking and sitting times after meals harmed health [11,12,13]. Voluntarily refraining from leaving the home has not only led to a decrease in physical activity but also to social isolation. A study in Spain found a correlation between social isolation during the COVID-19 self-isolation period and a reduced health status in older people [14]. In an Australian study on social activity, COVID-19 self-isolation affected social relationships and reduced life satisfaction, with over 95% of 3745 participants choosing to maintain a sufficient social distance during COVID-19 self-isolation [15]. As the social distance increased, housing became the main location for daily activities [16]. Before the COVID-19 crisis, 8.4% of people chose to work from home (WFH) [20]. However, after a state of emergency was declared, 34.6% of full-time employees chose to WFH [21]. During the COVID-19 self-isolation period, housing was the main location in which people lived and worked [22]. In a study by Andrea Amerio, in a small flat, the risk of developing depressive symptoms may be four times higher [23]. Guo et al. [24] found a correlation between apartments and mild depression. Most previous studies have focused on a single or a few daily activities, such as physical exercise and social interaction, to elucidate their relationship with health [5,6,7,8,9,10,11,12,13,14,15,16]. There is still a lack of knowledge on the impact of various daily activities on health status. Investigating multifaceted daily activities and health status would provide a better understanding of the correlation between daily activities and health status during COVID-19.

In addition, many previous studies have elucidated the relationship between housing type and health status, particularly mental health. However, there is still a lack of research capturing the correlations between daily activities, health status, and housing type. To date, several previous studies have shown a relationship between the living environment and health status, as well as daily activities [17,18]. These studies have clarified the relationship between the living environment and health status in housing complexes, while there is still a lack of research examining the relationship between the living environment and health status in other housing types, especially in detached houses. Japanese housing is classified according to the housing type into apartments, detached houses, and row houses. The proportions of detached houses, apartments, and row houses in Japan are 53.6%, 43.6%, and 2.6%, respectively. In the Osaka Prefecture, the proportions of detached houses, apartment buildings, and row houses are 40.7%, 55.4%, and 3.8%, respectively [25]. Investigating the daily activities and health status based on these housing types will provide findings corresponding to the housing types that contributed to the health maintenance of residents during the COVID-19 self-isolation period.

This study elucidates the relationship between daily activities and the health status of residents in a suburban residential area during COVID-19 self-isolation based on age and housing type. The relationship between the daily activities and the health status of residents in a suburban residential area with an aging population is investigated to identify the daily activities corresponding to each housing type that helped residents to maintain their health status.

## 2. Materials and Methods

A questionnaire survey was conducted on the residents of 1-chome and 5-chome of Teraikedai, Kongo Danchi District, Tondabayashi City, Osaka Prefecture. The questionnaire investigated how the COVID-19 pandemic affected daily activities and health status. The survey targeted residents aged 18 years or older, living in 1-chome and 5-chome of Teraikedai.

### 2.1. Survey Area

The Kongo District has a total area of approximately 216 ha. The Kongo District comprises Teraikedai, Kunokidai, and Takabedai in Tondabayashi City and Kongo in Osaka-Sayama City. In 2018, the Ministry of Land, Infrastructure, Transport and Tourism’s “Nationwide List of New Towns” identified the Kongo District as a suburban residential area [26]. Since the survey cohort was New Town, the Kongo District was targeted. In 2020, 227 households lived in detached houses, and 753 households lived in apartments in Teraikedai 1-chome. Moreover, 220 households lived in detached houses, and 337 households lived in apartments in Teraikedai 5-chome [27]. Among the households, this study performed an exhaustive questionnaire survey on residents who were associated with neighborhood associations in Teraikedai 1-chome and 5-chome. The percentage of residents living in apartments was 72.92%; as of August 2022, the population was 7817 and the aging rate was 36.54%, making it one of the largest aging populations in the Osaka Prefecture [28], Figure 1.

### 2.2. Questionnaire Survey

An exhaustive survey of 654 households among the Teraikedai 1-chome and 5-chome neighborhood associations was conducted between 16 March and 15 April 2022, by means of self-administered questionnaires, Table 1. Participants were consecutively enrolled after the study description and obtaining informed consent. Group leaders and street committee members distributed two questionnaires to each household and collected them. The contact details of the correspondent faculty and the volunteer group were provided on the questionnaires so that respondents could ask any question about which they were not sure. We gave participants adequate time, one month, to answer the questionnaire, because the method of administration and the time dedicated to each interviewee may influence the choice of answers [29]. Two questionnaires per household were distributed and collected via group leaders and street committee members. A total of 476 responses were received, with 427 valid responses, excluding 49 respondents who did not complete the SF-12v2 questionnaire. A breakdown of the respondents shows that there were higher numbers of women (246, 61.65%, mean ±SD: 47.89 ± 10.76), older people aged 65 years and above (227, 57.16%, mean ±SD: 76.15 ± 6.50), and residents of housing complexes (315, 73.77%). The survey period was from 16 March 2022 to 15 April 2022. The survey period corresponded to the implementation period of the third round of the new vaccination program. It was a period when the number of newly infected people in the Osaka Prefecture was around 5000 and was gradually decreasing.

### 2.3. Sociodemographic Characteristics and Housing Types

Age, gender, occupation, and housing types were collected as characteristics of the respondents. The respondents were categorized according to age into two groups: adult (under the age of 65) and elderly (over the age of 65). Respondents were categorized according to their characteristics into four types: detached house residents over the age of 65 (DROA), apartment residents over the age of 65 (AROA), detached house residents under the age of 65 (DRUA), and apartment residents under the age of 65 (ARUA).

### 2.4. Basic Survey on Social Life

The Basic Survey on Social Life is a survey conducted every five years by the Ministry of Internal Affairs as the government’s survey for policy making [30], which focuses on people’s daily activities. The purpose of the survey is to obtain basic data to clarify the actual status of people’s social life, such as the distribution of their living time and the main activities in their leisure time. According to the survey on time use and leisure activities in 2021 [30], the daily activities were classified into 18 categories: sports, exchange socializing, medical treatment, rest, housework, patrolling, meals, commuting to work and school, work, study, care and nursing, childcare, shopping, moving, television, personal development, recreational hobbies, and social participation activities. To complete the survey of daily activities, questions about sleep quality were added. This study followed this approach and developed the daily activities questionnaire with the following five levels: no, less than 1 h, 1~3 h, 4~6 h, and more than 6 h (Appendix A).

### 2.5. Health Indicators

To measure quality of life, the Short Form 12 Health Survey Version 2 (SF-12v2) instrument is frequently used in health assessments. The SF-12v2 is a simple questionnaire but with established reliability in Japan and abroad. The SF-12v2 is the abridged practical form of the SF-36. It is a widely used screening device for measuring physical, mental, and social well-being to assess the quality of life of respondents [31]. The SF-12v2 health score measure has 3~5 options for each item and is divided into eight indicators (physical functioning, role physical, bodily pain, general health, vitality, social functioning, role emotional, and mental health) [32]. This study used the SF-12v2 to measure the physical health scores (PCS) and mental health scores (MCS) [33]. SF-12v2 health scores take values ranging from 0 to 100, with 0 representing the lowest health level and 100 representing the highest health level. The final score can be compared with the average score of the US standard (50.12 for the PCS and 50.04 for the MCS) to determine the physical and mental health of the population.

### 2.6. Statistical Analysis

Before the analysis, the normality of each quantitative variable was checked. The SF-12v2 results were expressed as PCS and MCS for the continuous variables. Age (including over 65 years old and under 65 years old) and housing types (detached house and apartment) were treated as nominal variables. The statistically significant difference in the residents’ health scores corresponding to age and housing type were examined using the Tukey–Kramer HSD test for continuous and nominal variables.

Regression models were performed on a data set of nineteen categories of daily activities. Forward–backward stepwise selection methods were used, with an F probability of 0.05 for entry and 0.05 for removal. The explanatory variables were the nineteen daily activities of the Basic Social Life Survey, and the objective variables were the physical and mental health scores. JMP (JMP Pro Version 16.0.0, SAS Institute, Inc., Cary, NC, USA) was used for the analysis.

## 3. Results

### 3.1. Participant Characteristics

The baseline characteristics of all participants in each housing type are shown in Table 2. Response data were analyzed in four categories: DROA, AROA, DRUA, and ARUA. The majority of elderly residents lived with their families (273, 86.7%), while residents under 65 lived alone (14, 8.3%) or with their spouses (34, 20.1%).

### 3.2. Health Score

The health scores of respondents are shown in Figure 2. Respondents had an average PCS of 48.56 and an average MCS of 46.95. The mean PCS and MCS of the residents were both lower than the standard health points of the SF-12v2 (50.12 for PCS and 50.04 for MCS) [30]. On the other hand, the DRUA had a PCS of 52.9, above the average PCS. Regarding MCS, the DRUA also had the highest health score of the four groups at 48.04, but this was below the US average score. The MCS for residents of ARUA was even lower, at 43.14.

The Tukey–Kramer HSD test results are presented in Table 2. For PCS, significant differences at the 1% level between the DROA and AROA were analyzed by the Tukey–Kramer HSD. The Tukey–Kramer HSD also analyzed significant differences at the 1% level between DRUA and ARUA. For MCS, significant differences were found at the 5% level in DRUA and ARUA.

The PCS and MCS (Figure 3) were analyzed independently for age and housing type, with the following two findings. Firstly, among the same age groups, residents of detached houses had higher PCS and MCS than residents of apartments. Secondly, for the same housing type, PCS and MCS were lower for residents over the age of 65 than for residents under the age of 65. There was one exception: the AROA had a higher MCS than the ARUA.

### 3.3. Relationship between Daily Activities and Health Status

#### 3.3.1. Physical Health Score

The relationship between daily activities and PCS by resident age and housing type is shown in Table 3. For the DROA, work, recreational hobbies, and sports were correlated with higher PCS. For the AROA, exchange socializing was correlated with higher PCS. For the residents under the age of 65, medical treatment was the most important factor in lowering PCS, and for the DRUA, sports and social participation activities were correlated with PCS. Thus, the DROA and AROA are commonly correlated with sports. Furthermore, the DRUA and ARUA are commonly correlated with medical treatment.

#### 3.3.2. Mental Health Scores

The relationship between daily activities and MCS by resident age and housing type is shown in Table 4. For the DROA, the nineteen daily activities’ data were imported into the stepwise validation (F-test). The model did not undergo significant changes and the variables were removed from Table 4. For the DRUA, sports were correlated with higher MCS. For the ARUA, recreational hobbies were correlated with higher MCS.

## 4. Discussion

This study found that there was no significant difference in health scores between the DROA and AROA, but there was a significant difference in health scores between the DRUA and ARUA. Then, this study analyzed the factors that led to the significant difference in health scores among residents aged under 65 according to the housing type in terms of the relationship between daily activities and health scores. It was found that for residents of detached houses, physical activity was associated with physical health. For residents of apartments, communication and hobbies were associated with mental health. This conclusion is a novelty because the analysis clarified the factors related to health scores according to not only age but also housing type.

The results of this paper are described below. It was found that during COVID-19 self-isolation, most of the residents’ PCS and MCS in suburban residential areas were below the US standard (50.12 for the PCS and 50.04 for the MCS). The relationship between the health scores of residents in suburban residential areas and nineteen daily activities, and between the respective health scores and daily activities, was then clarified. Of these, no statistically significant differences were found at the 5% level in either the PCS or MCS between the DROA and AROA. On the other hand, there were statistically significant differences at the 5% level in both PCS and MCS between the DRUA and ARUA. The health scores of the DRUA were significantly better than those of the ARUA. Regarding PCS, the DRUA and ARUA correlated mainly with exchange socializing, work, recreational hobbies, and sports in terms of daily activities. From this, it can be inferred that differences in the health scores of residents aged under 65 are correlated with the amount of time spent on sports and exchange socializing. In terms of MCS, the daily activities with different regression coefficients for the DRUA and ARUA were concentrated in sports and recreational hobbies.

Several previous studies have shown that, during COVID-19, residents living in detached houses had lower reductions in physical activity compared to those living in apartments [34,35]. During COVID-19, residents living in detached houses had more opportunities to play sports or perform gardening or domestic work [36]. Cerin, E’s study produced the same results, with residents of detached houses being more physically active [37]. The area has a building coverage ratio of 50% and a building setback of 1 m due to a building agreement, so there is relatively ample space for the site and garden area of the detached house. The total floor area of apartments tends to be much smaller than that of detached houses. The limitation of the living area may reduce the activity of the residents. In Teraikedai, instead of outdoor exercise, indoor and in-garden exercise might have caused the higher health scores for the DRUA and DROA.

In studies by Yomoda, residents of detached houses, which have more space compared to apartments, experienced lower declines in physical activity during COVID-19 due to exercising indoors [38]. In Nurul Habib’s study, residents living in detached homes would return home earlier and be less active outside the house [39]. In this study of detached house residents, sports had a higher correlation coefficient with health, perhaps associated with lower activity levels outside and higher levels of indoor sports.

In several previous studies, poor housing conditions negatively affected the health status of residents [23,40,41], especially for apartments, where problems such as overcrowding and poor sound insulation profoundly affect people’s daily lives and health status [42]. During the COVID-19 pandemic, more residents chose to work from home [21], and spent more time at home. Residents who worked from home were more likely to suffer from noise and get stressed [43]. In contrast, apartments which have more crowded housing environments were more likely to be negatively affected by working from home compared to detached houses. This may explain the better health status of the DRUA compared to the ARUA.

In a study by Midway, rest and recreational hobbies were important factors for maintaining well-being during COVID-19 [44]. To cope with COVID-19 self-isolation, recreational hobbies could increase creativity in individuals’ daily lives [45,46]. This study proved that recreational hobbies are an essential factor for high PCS among DRUA and proved that recreational hobbies are an essential factor for high MCS among ARUA. This suggests that the lower MCS among the ARUA may be related to reduced recreational hobbies.

This had an impact on residents’ daily activities during COVID-19 self-isolation. In this study, the mean scores of the PCS and MCS in the DROA, AROA, DRUA, and ARUA were 46.92, 44.3, 53.07, and 49.64, respectively, while the mean scores of the MCS were 47.83, 47.29, 48.57, and 44.94. These were lower than the standardized mean of 50.12 for the PCS and 50.04 for the MCS [36]. It is estimated that COVID-19 self-isolation was a factor in reducing both PCS and MCS. In contrast, among two Saudi studies, one by Aisha Alhofaian et al. found a value of 44.7 for PCS of 44.7 and 34.6 for MCS [47], while the study by Manal Mohammed Hawash et al. found a value of 41.65 for PCS and 32.34 for MCS [48]. Compared to the Saudi survey data, the health scores of the residents of Terakhedai were relatively high.

This study has the following limitations. First, the study is based on a self-administered survey. Therefore, the data obtained in this study are subjective evaluations. Second, the study is based on a cross-sectional design and descriptive statistics. Therefore, this study does not elucidate the causal relationships. For future research, longitudinal studies need to be conducted to discuss the impact of each daily activity on the health status of the residents.

## 5. Conclusions

In conclusion, this study found no statistically significant difference in health scores between DROA and AROA, but a statistically significant difference in health scores between DRUA and ARUA was found. The health scores of the DRUA were significantly better than those of ARUA. Factors contributing to the differences were assumed to be related to time spent on sports and recreational hobbies. In addition, sports had a significant total effect on physical health, and recreational hobbies significantly affected mental health. This study also found that the physical and mental health scores of residents in suburban residential areas decreased during the COVID-19 self-isolation period.

## Figures and Tables

**Figure 1 ijerph-20-02639-f001:**
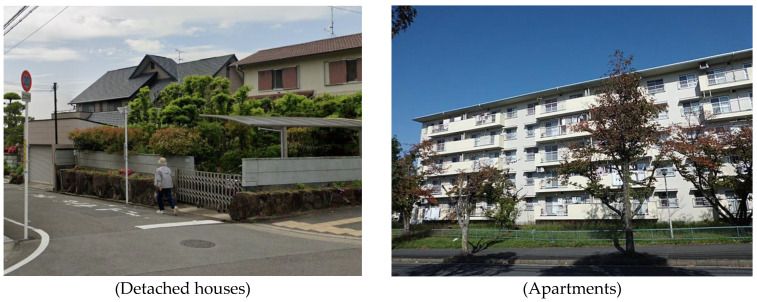
Examples of each housing type.

**Figure 2 ijerph-20-02639-f002:**
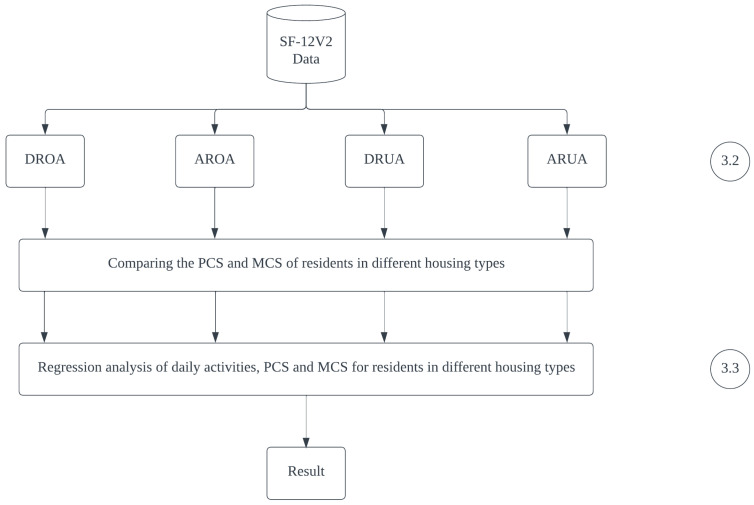
Processing of questionnaire date.

**Figure 3 ijerph-20-02639-f003:**
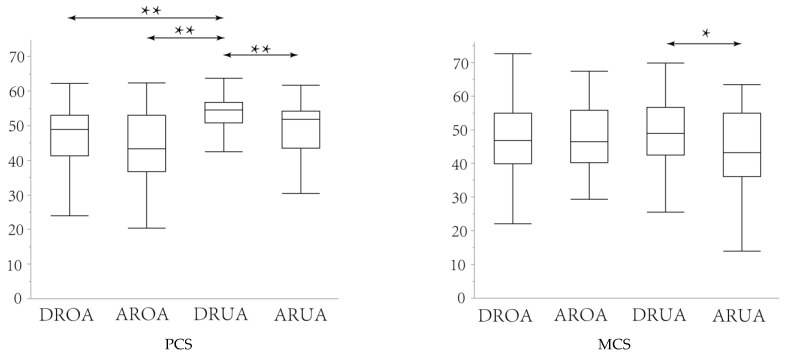
PCS and MCS (* *p* < 0.05, ** *p* < 0.01).

**Table 1 ijerph-20-02639-t001:** Distribution and collection of questionnaires.

Number of questionnaires distributed	1308
Number of questionnaires collected	476
Research location	Kongodanchi Teraikeidai, Tondabayashi, Osaka, Japan
Method of investigation	Exhaustive questionnaire
Research period	16 March 2022–15 April 2022
Number of valid questionnaires	427
Valid availability ratio	32.6%
Respondents’ age distribution	Under 65 years: 200 (46.84%) Over 65 years: 227 (53.16%)
Respondents’ sex distribution	Male: 153 (35.83%) Female: 246 (57.61%) Non-respondents: 28 (6.56%)

**Table 2 ijerph-20-02639-t002:** Characteristics of residents.

	DRUA	ARUA	DROA	AROA
Gender				
Male	73 (17.10%)	19 (4.45%)	42 (9.84%)	19 (4.45%)
Female	85 (19.91%)	32 (7.49%)	96 (22.48%)	33 (7.73%)
Unknown	12 (2.81%)	6 (1.26%)	7 (1.64%)	3 (0.7%)
Family structure				
Living alone	29 (6.79%)	12 (2.81%)	1 (0.23%)	2 (0.47%)
Couple	95 (22.25%)	26 (6.09%)	28 (6.56%)	8 (1.87%)
Nuclear family generations	35 (8.2%)	17 (3.98%)	100 (23.42%)	41 (9.6%)
Three generations	4 (0.94%)	1 (0.23%)	11 (2.58%)	1 (0.23%)
Other	7 (1.64%)	1 (0.23%)	5 (1.17%)	3 (0.7%)
Occupation				
Company employees and civil servants	10 (2.34%)	2 (0.47%)	66 (15.46%)	23 (5.39%)
Self-employed profession	15 (3.51%)	2 (0.47%)	13 (3.04%)	5 (1.17%)
Household chores	67 (15.69%)	18 (4.22%)	26 (6.09%)	9 (2.11%)
Student	0	0	2 (0.47%)	0
Part-time job	9 (2.11%)	5 (1.17%)	30 (7.03%)	9 (2.11%)
Fixed year retirees	55 (12.88%)	23 (5.39%)	4 (0.94%)	1 (0.23%)
Other	14 (3.28%)	6 (1.41%)	4 (0.94%)	8 (1.87%)
Unknown	0	1 (0.23%)	0	0

**Table 3 ijerph-20-02639-t003:** Logistic regression results of the PCS.

		B	Bate	SE	t	*p*	95%CI		VIF
							Lower	Upper	
DROA	(Constant)	27.17	0	4.84	5.62	0.0001 **	17.59	36.75	
	Meals	1.74	0.10	1.50	1.16	0.25	−1.22	4.70	1.04
	Work	1.69	0.24	0.59	2.86	0.0049 **	0.52	2.86	1.07
	Recreational hobbies	3.04	0.25	1.09	2.80	0.0059 **	0.89	5.19	1.23
	Sports	2.48	0.21	1.02	2.43	0.0167 *	0.46	4.51	1.17
AROA	(Constant)	41.26	0	4.94	8.36	0.0001 **	31.33	51.20	
	Exchange socializing	4.50	0.37	1.58	2.85	0.0065 **	1.32	7.68	1.00
	Medical treatment	−3.4	−0.23	1.98	−1.72	0.09	−7.40	0.57	1.00
DRUA	(Constant)	56.91	0	2.04	27.84	0.001 **	52.86	60.95	
	Sports	2.17	0.22	0.82	2.66	0.0088 **	0.56	3.80	1.06
	Social participation activities	−2.49	−0.18	1.18	−2.11	0.0369	−4.82	−0.15	1.09
	Medical treatment	−3.24	−0.24	−1.11	−2.93	0.0040 **	−5.43	−1.05	1.05
ARUA	(Constant)	56.27	0	3.02	18.66	0.0001 **	50.21	62.33	
	Medical treatment	−5.64	−0.41	1.88	−3.00	0.0043 **	−9.43	−1.86	1.03

Note: DROA: detached house residents over the age of 65, AROA: apartment residents over the age of 65, DRUA: detached house residents under the age of 65, ARUA: apartment residents under the age of 65, * *p* < 0.05, ** *p* < 0.01.

**Table 4 ijerph-20-02639-t004:** Logistic regression results of the MCS.

		B	Bate	SE	t	*p*	95%CI		VIF
							Lower	Upper	
AROA	(Constant)	39.48	0	2.90	13.62	0.0001 **	33.66	45.29	
	Exchange socializing	4.53	0.39	1.49	3.03	0.0038 **	1.53	7.52	1.00
DRUA	(Constant)	44.43	0	1.97	22.50	0.0001 **	40.53	48.34	
	Sports	2.53	0.17	1.21	2.09	0.038 *	0.14	4.90	1.00
ARUA	(Constant)	40.03	0	3.98	10.06	0.0001 **	32.04	48.02	
	Recreational hobbies	3.42	0.34	1.46	2.34	0.0235 *	0.48	6.36	1.13
	Exchange socializing	−2.37	−0.18	1.89	−1.25	0.2163	−6.18	1.44	1.13

Note: The nineteen daily activities’ data from the DROA were imported into the stepwise validation (F-test). The model did not undergo significant changes and the variables were removed, DROA: detached house residents over the age of 65, AROA: apartment residents over the age of 65, DRUA: detached house residents under the age of 65, ARUA: apartment residents under the age of 65, * *p* < 0.05, ** *p* < 0.01.

## Data Availability

The data presented in this study are available from the corresponding authors.

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
