# Peer review of "Relationship between Health Status and Daily Activities Based on Housing Type among Suburban Residents during COVID-19 Self-Isolation"

_ijerph, 2023, doi:10.3390/ijerph20032639_

Round 1
Reviewer 1 Report
The article investigates the relationship among health status, daily activities, and housing types for a sample of 378 residents of 1-chome and 5-chome of 122 Teraikedai, in Osaka Prefecture, during the COVID-19 self-isolation period.
The article’s main strength is the clear definition of the research question, i.e. determining whether the relationship between changes in daily activities and health status during COVID-19 self-isolation depended not only on age but also on housing type. Also, the article clearly identifies two specific knowledge gaps, which addresses in the study.
The article’s topic is relevant to the Journal’s scope and specifically to the Special Issue (“The Impact of Built, Natural, Social, and Virtual Environments on Human Health”).
Overall, the manuscript is well structured and gives a proper description of the main theme, the topic’s context, and research findings. The cited references are relevant and, with very few exceptions, up-to-date, and the Literature review section gives an adequate analysis of the research background.
The research design is appropriate for the set objective; the methodology used is described in its main features and scientifically sound, and the obtained results are consequently reproducible. Nevertheless, materials, methods, and results deserve some more clarifying effort, for the article to be fully understandable and clear enough for readers:
- the decision to select respondents among neighborhood associations should be clarified and made explicit, in subsection 1.2 or in subsection 2.1, as it may give rise to questions about whether the sample is sufficiently representative of the case-study area or, conversely, some unknown peculiarities of associations’ communities may have any influence
- the article mentions only incidentally the use of two questionnaires (line 144), but these are not described: an “SF-12v2 questionnaire” (Line 147) and a “daily activities questionnaire” (line 168) are mentioned, which remain rather obscure. I suggest that the two questionnaires be described, explaining their purpose and content; the list of questions contained in the two questionnaires (e.g., as supplementary material) would also allow a more understandable reading of the work
- some subsections in the Materials and Methods section should be improved and better clarified. Particularly in Subsection 2.5 (Health indicators), acronyms should be extensively defined (e.g., “SF-12v2” among others) and indicators’ meaning and content should be better and more extensively described, as they remain rather obscure and uneasy to interpret
- lines 83-85: the cited reference ([13]) actually refers to a study for Spain, not Italy
- lines 202-207: the text is a bit unclear (see “Respondents had an average PCS of 48.56 and an average MCS of 46.95, below the average PCS and MCS….”) and should be reformulated
- subsections 3.3.1 and 3.3.2 are really unclear, due to the lack of explanation of the indicators mentioned in Tables 3 and 4.
The Discussion section is appropriate and consistent in general, but lines 246-250 contain rather cryptic statements, as they do not actually explain differences between housing types as announced in lines 244-246.
Conclusions correctly identify the limitations of the study, but they could also be improved by identifying open research areas and expanding considerations on the usefulness of the results.
Due to the generally excessive briefness of the work in the mentioned points, I suggest integrating more generous descriptions in the related subsections, in order to make the work really good and suitable for publication.
Author Response
Dear Reviewer:
We appreciate the reviewer for the generous comment on the manuscript. We have attached our response letter in PDF format. We believe that the manuscript is now suitable for publication in International Journal of Environment Research and Pulish Health and look forward to hearing from you concerning your decision.
Yours sincerely

Reviewer 2 Report
The subject is interesting and the manuscript is well structured and organised. A good overview of the literature is presented. The experimental design is correct as well as data analysis.
The manuscript can be published in the present form.
Author Response
Dear Reviewer:
We appreciate the reviewer for the generous comment on the manuscript. We have attached our response letter in PDF format. We believe that the manuscript is now suitable for publication in International Journal of Environment Research and Publish Health and look forward to hearing from you concerning your decision.
Yours sincerely

Reviewer 3 Report
Thank you for asking me to view this article.
The subject under study is very interesting especially in relation to the fact that given the COVID-19 pandemic, the introduction of emergency measures (especially in the period prior to the mass vaccination campaign against SARS-CoV-2) such as social distancing and forced isolation imposed in order to reduce infections have had an impact on people's health. Working habits, relationships and emotional ties in the lives of many individuals (also in relation to housing conditions) have undergone profound changes, sometimes causing a sense of disorientation and discouragement, therefore investigating the impact of these aspects on lifestyle habits, health and behavior is of relevance to the scientific community.
However, despite the topicality of the topic explored by the authors, Major revision is considered necessary before proceeding with a further revision process.
Here are my suggestions in this regard:
I think the introductory section described itself in a scattered and cluttered way. In the first paragraph, the authors refer to the measures imposed for the containment of infections but do not describe the effects that this precaution could have caused on the population with examples that help the reader understand the context in which the research question developed. What is the reference cohort? Is there evidence describing the impact of the lockdown in the reference cohort? The reference bibliography that accompanies the contents of the introduction is rich but should serve to complete the text, not to replace the contents that are not incomplete. The second paragraph, in my opinion, does not offer any additional information compared to the previous paragraph, in which the authors repeatedly reiterate the purpose of the work, making the concept redundant. I suggest removing it and using lines 60-67 in a paragraph dedicated to "study setting" in the methods section.
The term “epidemic” and the term “pandemic” refer to different epidemiological situations and therefore should not be used as synonyms in a manuscript. The authors are invited to replace the term "epidemic" with "pandemic" when referring to the COVID-19 pandemic.
Paragraph 1.3 refers to a Literature Review but the manuscript was submitted to the journal as an “Article”; therefore the literature review must be implicitly conducted for the drafting of a manuscript. I suggest the authors to use the contents of this paragraph to describe the background. In this way the article would be presented in a much more coherent way, making the reading smoother.
Line 85. The authors refer to an Italian study (reference 13) while the quotation refers to a study conducted in Spain. Please check.
The authors speak of the administration of a questionnaire but it is not clear the method of administration nor the method of recruiting the families or the ways in which they have expressed the choice to join. Furthermore, for the purposes of reproducibility of the study in other contexts as well, it is important to include the questionnaire with the manuscript (also as supplementary material) and to describe the data collection phase more accurately. In fact, it is well known that the recruitment methods of the interviewees, the consequentiality of the questions, the method of administration and the time dedicated to each interviewee (or groups of interviewees such as families) are of particular importance and could also influence the choice of answers (doi: 10.3390/ijerph191811212). Referring the questionnaire to a study that has already been conducted is correct but, in my opinion, it is important to deepen its contents by adapting them to the context in which the research question develops. Specifically, one of the questionnaires is well known in Japan, but not so in the rest of the world. The authors must make the manuscript fluent for the reader, providing methodological aspects that seem implicit to them.
I suggest the authors to review the methods section.
The contents presented in the discussion are partly a repetition of what has already been described in the results. The authors reaffirm the originality of the study on the basis of the association found between age, housing conditions and health during the lockdown following the COVID-19 Pandemic. The academic literature describes that housing can be a key determinant of health, indeed it has been estimated that inadequate housing conditions (eg, overcrowding) can cause unhealthy conditions (doi: 10.1155/2020/7642658); it is plausible that this is amplified by a condition of isolation imposed also in relation to the age of the individuals and considering the results of the survey conducted by the authors, I suggest commenting on this aspect as well so as to open food for thought and broaden their empirical value, that is, the contribution they offer to the theoretical and empirical progress of the research conducted.
Minor revisions. The abstract should briefly summarize the context in which the research question is developed, the methodology chosen to answer the question, the results obtained and a brief conclusion. The description of the contents, therefore, should faithfully summarize the sections of which the manuscript is composed. What the reader deduces from the abstract is only partially consistent with what is explored in the introduction and methods. For example, the abstract does not clarify in which country the study is being conducted but generic reference is made to "suburban residential". It is suggested to review the abstract by making the background more concise and to organize its contents so that they are consistent with the text, including the survey methodology adopted to answer the research question.
Author Response

(The authors gave the same response as above.)

Round 2
Reviewer 1 Report
The revised version of the article addressed most of the issues raised, except for Comment No.1:
“The decision to select respondents among neighborhood associations should be clarified and made explicit”.
Although the authors introduced reference that explain “why the survey area is representative of suburban residential area”, their decision to focus only on neighbourhood associations still remains unmotivated, which was my point actually.
By adding these last motivations, which are still unaddressed, the article will be suitable for publication; that is why my final overall recommendation is "Accept after minor revision".
Author Response
Dear Reviewer:
We appreciate the reviewer for the generous comment on the manuscript. We have attached our response letter in PDF format. We believe that the manuscript is now suitable for publication in International Journal of Environment and Public Health and look forward to hearing from you concerning your decision.
Yours sincerely
